# Self-supervised pseudo-colorizing of masked cells

**Royden Wagner©\*, Carlos Fernandez Lopez, Christoph Stiller**

Karlsruhe Institute of Technology (KIT), Karlsruhe, BW, Germany

\* royden.wagner@kit.edu

**Data Availability Statement:** All used datasets are available from http://celltrackingchallenge.net/datasets/.

**Funding:** The author(s) received no specific funding for this work.

## Abstract

Self-supervised learning, which is strikingly referred to as the dark matter of intelligence, is gaining more attention in biomedical applications of deep learning. In this work, we introduce a novel self-supervision objective for the analysis of cells in biomedical microscopy images. We propose training deep learning models to pseudo-colorize masked cells. We use a physics-informed pseudo-spectral colormap that is well suited for colorizing cell topology. Our experiments reveal that approximating semantic segmentation by pseudo-colorization is beneficial for subsequent fine-tuning on cell detection. Inspired by the recent success of masked image modeling, we additionally mask out cell parts and train to reconstruct these parts to further enrich the learned representations. We compare our pre-training method with self-supervised frameworks including contrastive learning (SimCLR), masked autoencoders (MAEs), and edge-based self-supervision. We build upon our previous work and train hybrid models for cell detection, which contain both convolutional and vision transformer modules. Our pre-training method can outperform SimCLR, MAE-like masked image modeling, and edge-based self-supervision when pre-training on a diverse set of six fluorescence microscopy datasets. Code is available at: https://github.com/roydenwa/pseudo-colorize-masked-cells.

## Introduction

The ambitious goal of deep learning research is to develop intelligent generalist models that can solve a wide variety of tasks. Supervised learning with massive amounts of labeled data does not scale to the complexity of this goal. In self-supervised learning, however, supervisory signals are generated from unlabeled data, making it more scalable. Therefore, self-supervised learning, which is strikingly referred to as the dark matter of intelligence [1], is gaining more attention in many applications of deep learning. Especially in biomedical applications, where labeling data often requires support of trained experts, self-supervised learning can accelerate research progress and reduce costs [2]. In this work, we introduce a novel self-supervision objective for the analysis of cells in biomedical microscopy images. In contrast to recent methods for self-supervised learning on cell images (e.g., [3, 4]), we do not use contrastive learning [5], but propose a unique self-supervision objective tailored to cell images. We propose training deep learning models to pseudo-colorize masked cells. We use a physics-informed pseudo-spectral colormap that is well suited for colorizing cell topology. Compared to a spectral

**Competing interests:** The authors have declared that no competing interests exist.

colormap, the used colormap has a higher color variance for low intensity levels. Therefore, this pseudo-spectral colormap can better highlight cell nuclei and their surroundings in areas of low intensity and low contrast in microscopy images. Inspired by the recent success of masked image modeling (e.g., [6, 7]), we additionally mask out cell parts to increase the complexity of the objective and further enrich the learned representations. Furthermore, we build upon our previous work [8] and train hybrid models for cell detection, which contain both convolutional and vision transformer modules. The convolutional modules are used to capture local information, while the vision transformer modules are used to capture global information. Overall, the contributions of our work are twofold:

1. We propose a novel self-supervision objective for the analysis of biomedical cell images that combines pseudo-colorizing and masked image modeling.

2. We use a recent deep learning model for cell detection, which contains both convolutional and vision transformer modules, to evaluate the proposed self-supervision objective.

## Related work

### Self-supervised learning on biomedical cell images

Ciga et al. [3] and Perakis et al. [4] use contrastive learning and build upon SimCLR [9] to train deep learning models in a self-supervised manner on cell images. In contrastive learning for visual representations, the learning objective is to maximize agreement between two different augmented views of a single sample, while using the remaining samples in a batch as negative examples. Experiments show that large batch sizes and correspondingly huge datasets and computational resources are needed to maximize performance (e.g., a batch size of 32k for SimCLR on ImageNet). Dmitrenko et al. [10] show that a small convolutional autoencoder (only 190k parameters) trained on cell images can outperform much larger general purpose pre-trained models (e.g., ViT-B/8 [11], 85M parameters) on classifying drug effects. This demonstrates the potential of autoencoding as pre-training on cell images. However, recent work on autoencoders (e.g., [6, 12]) suggests that masking parts of the input further enriches the learned representations. Kobayashi et al. [13] propose protein identification as self-supervision objective for pre-training on biomedical cell images. While being well tailored to cell images, this approach requires basic annotations of protein IDs. Dawoud et al. [14] use edge detection as self-supervision objective to pre-train deep learning models for cell segmentation. Edge detection is closely related to accurate segmentation of cell borders, but is a fairly simple task which could lead to less expressive learned features.

### Colorizing as self-supervision objective

Zhang et al. [15] convert natural color images to grayscale and use the recolorization to the CIE lab color space as self-supervision objective. Vondrick et al. [16] leverage the temporal coherency of color in natural videos to learn colorizing following frames based on a reference frame. Both approaches are promising for natural images or videos, but exploit color-related features that are not present in biomedical grayscale microscopy images.

## Materials and methods

### Pseudo-colorize masked cells as self-supervision objective

Autoencoders [17] are classical deep learning models for self-supervised representation learning. During training an autoencoder learns to map its input to a latent representation and to

reconstruct the input from the latent representation. The general architecture of an autoencoder contains a contracting part called encoder and an expanding part for reconstruction called decoder. Popular applications include compression, where the learned latent representation is smaller than the input, or denoising [18], where input signals are corrupted by noise and uncorrupted signals are reconstructed. In this work, we use a new form of autoencoding by training to reconstruct pseudo-colorized versions of input images. Specifically, we train deep learning models to pseudo-colorize cell images as self-supervised pre-training for cell detection. As previously shown [10], basic autoencoding of cell images can improve the performance on downstream tasks such as drug effect classification. We argue that reconstruting a pseudo-colorized version is more difficult than reconstructing the original input image. Thus, it can lead to more expressive learned features and further improve the performance on downstream tasks. Fig 1(a) shows pseudo-colorized cell images and the characteristics of the used colormaps. We are using colormaps provided by the matplotlib library [19]. The grayscale plots of the colormaps are generated by computing the perceived brightness $B_P$ from RGB values using the HSP color system [20], as follows:

$$B_P = \sqrt{0.299R^2 + 0.587G^2 + 0.114B^2} \tag{1}$$

We select four colormaps from different colormap categories to cover a wide range of colormaps in the following. The `rainbow` colormap is a physics-informed spectral colormap based on the visible spectrum of light. The lowest intensity levels are mapped to violette, the highest intensity levels to red. The perceived brightness gradually decreases from medium intensity levels towards low and high intensity levels. The `seismic` colormap is a

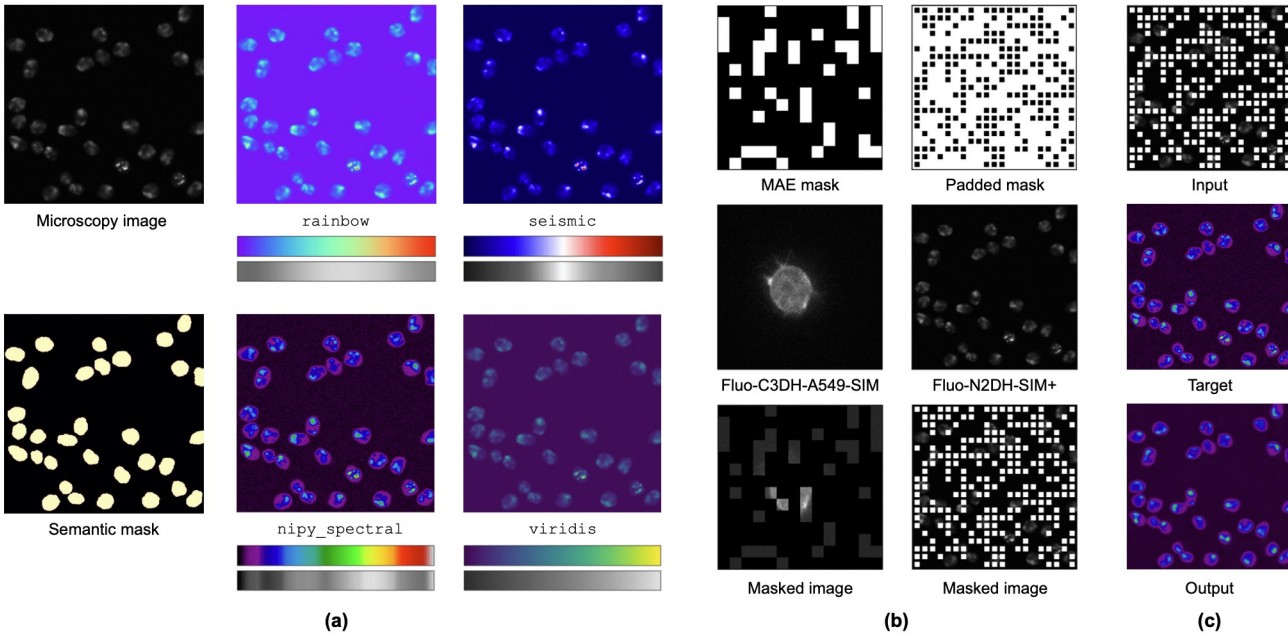

**Fig 1.** **(a)** Pseudo-colorization of fluorescence microscopy images and the corresponding colormaps. **(b)** Masking schemes and masked fluorescence microscopy images. MAE [6] masks cover 75% of images, whereas our proposed padded masks contain smaller patches and cover 33%. Image areas masked by our padded masking scheme are highlighted in white here to enhance their visibility. During pre-training, these areas are set to zero. **(c)** Proposed pre-training objective: Pseudo-colorize masked cells.

monotonically diverging colormap composed of two colors, blue and red. The perceived brightness rapidly decreases from medium intensity levels towards low and high intensity levels. The `nipy_spectral` colormap is a physics-informed pseudo-spectral color map that extends spectral colormaps by prepending black for low intensity levels and appending gray for high intensity levels. Therefore, this colormap has a higher color variance than spectral colormaps. Furthermore, the color transitions in the colormap are sharper than in spectral colormaps. This is further expressed in a less uniform perceived brightness. `viridis` is a perceptually uniform sequential colormap composed of violette, blue, green, and yellow. It contains smooth color transitions and the perceived brightness is gradually increasing from low intensity levels towards high intensity levels.

We propose to choose colormaps for pseudo-color autoencoding, where the generated color channels approximately match the semantics of the data. We hypothesise that approximating semantic segmentation by pseudo-colorization is beneficial for self-supervised pre-training on images. For data such as fluorescence microscopy images, where semantics are closely related to grayscale intensity levels, spectral or pseudo-spectral colormaps are suitable. Such colormaps map low intensities primarily to bluish colors, medium intensities to greenish colors and high intensities to reddish colors. Thus, the intensity levels are roughly divided into the three color channels in the RGB scheme and semantic segmentation is approximated. As shown in Fig 1(a), the semantic cell mask covers primarily areas of low intensity. Therefore, the pseudo-spectral `nipy_spetral` colormap, which has a higher color variance for such intensity levels, can better highlight cells in this context than the spectral `rainbow` colormap. In detail, cell bodies are primarily colorized in violette, which maps to the blue and red RGB channels, and cell nuclei are primarily colorized in blue and green.

Recent work on autoencoding as pre-training on images (e.g., [6, 7, 12]), suggests that masked image modeling can lead to more expressive learned features than the classical reconstruction objective. In masked image modeling, images are divided into non-overlapping patches and during pre-training these patches are randomly masked out. Thereby, the objective for an autoencoder becomes to reconstruct seen and unseen parts of an image. The idea is that during pre-training an autoencoder builds up internal representations for certain object classes and learns to leverage these to reconstruct masked image parts. Masked autoencoder (MAE) [6] is an autoencoder model with an encoder and decoder based on the vision transformer architecture [21]. Therefore, images of size $224 \times 224$ pixels are divided into non-overlapping $16 \times 16$ pixel patches. For an image size of $384 \times 384$ pixels, this corresponds approximately to a patch size of $27 \times 27$ pixels. During pre-training, 75% of these patches are randomly masked out. We argue that such a masking scheme is primarily designed for data where the objective is to identify one concept per image and not to localize multiple objects within an image. As shown in Fig 1(b), the MAE-like masking would mask out a large number of cells in a Fluo-N2DH-SIM+ microscopy image. Therefore, an autoencoder can not use partially visible cell parts as clues to reconstruct adjacent masked patches, but rather has to learn a representation for all cells combined in an image to reconstruct masked patches. In Fig 1, the MAE-like masking is therefore applied to a Fluo-C3DH-A549-SIM microscopy image, which only contains one cell. We argue that learning to leverage local or semi-local visual clues is beneficial for downstream tasks such as object detection or segmentation. Thus, we propose an alternate masking scheme for cell detection. Our padded masking scheme is composed of smaller patches ($12 \times 12$ pixels for an image size of $384 \times 384$ pixels) and all patches are padded with a padding size of 1/4 of the patch width. This surrounds all patches with a non-masked area and prevents large connected masked areas, which could mask out whole cells. The patches are masked out randomly with a probability of 50% using a discrete uniform distribution $U\{0, 1\}$ to choose the common pixel value of a patch. Considering the non-masked

padding areas the overall mean masking ratio becomes $33.\overline{3}\%$. In the following we combine the pseudo-color autoencoding with masked inputs and thus train deep learning models to pseudo-colorize masked cells.

## Model architecture

The comparison of vision transformers (ViTs) [21] and convolutional neural networks (CNNs) [22] in computer vision applications reveals that their receptive fields are fundamentally different [23]. The receptive fields of ViTs capture local and global information in both earlier and later layers. The receptive fields of CNNs, on the other hand, initially capture local information and gradually grow to capture global information in later layers. Therefore, we use MobileViT blocks [24] in the neck part of our proposed model to enhance global information compared to a fully convolutional neck part. Fig 2(a) shows the proposed model architecture.

We represent cells by their centroid, their width, and their height. Our model contains two fully convolutional heads to predict these cell properties. The first head predicts a heatmap for cell centroids, and the second head predicts the cell dimensions (width and height) at the position of the corresponding cell centroid. The heatmaps for cell centroids are generated by first creating a semantic map where cells are approximated by ellipses. Afterwards, we smooth each cell ellipse using a normalized box filter with a kernel size $k$ that is scaled to the corresponding cell height $h$ and width $w$:

$$k = (w//1.5, h//1.5), \tag{2}$$

where // represents integer division. Cell height and width are encoded in rectangles sized to 50% of the corresponding cell dimensions. These rectangles contain height and width values scaled relative to the input image size of $384 \times 384$ pixels.

## Prepended ContextBlock

The ContextBlock is a proposed extension to our CellCentroidFormer model. It serves two purposes: first, to encode spatial context from adjacent slices when analyzing 3D inputs slice-wise. Second, to provide our model with a unified input form of 3 views for 2D and 3D input data to jointly train on both types of datasets. As previously shown (e.g., [26, 27]), using tripletts of adjacent slices when analyzing 3D microscopy images of cells slice-wise can improve cell detection and segmentation. Fig 2(b) shows how we decompose a 3D microscopy images into tripletts of adjacent slices. In the ContextBlock, highlevel features are first extracted from adjacent slices $S$ using convolutional layers. These features are then merged with the current slice using a multiply-accumulate operation to generate a context tensor $T_{\text{context}}$ with three channels.

$$T_{\text{context}}(z) = \begin{bmatrix} \sigma(\text{Conv1x1}(\text{Conv3x3}(S(z-1)))) \cdot S(z) + S(z) \\ S(z) \\ \sigma(\text{Conv1x1}(\text{Conv3x3}(S(z+1)))) \cdot S(z) + S(z) \end{bmatrix}, \tag{3}$$

where Conv3x3 and Conv1x1 represent 2D convolutional layers with kernel sizes of $3 \times 3$ and $1 \times 1$ and $\sigma$ represents a sigmoid activation function. A Conv3x3 layer with 10 filters is used to detect high-level features, afterwards these features are fused along the depth axis using a Conv1x1 layer with one filter. We merge context from the previous slice $S(z-1)$ in the first channel, the second channel contains the current slice $S(z)$, and context from the next slice

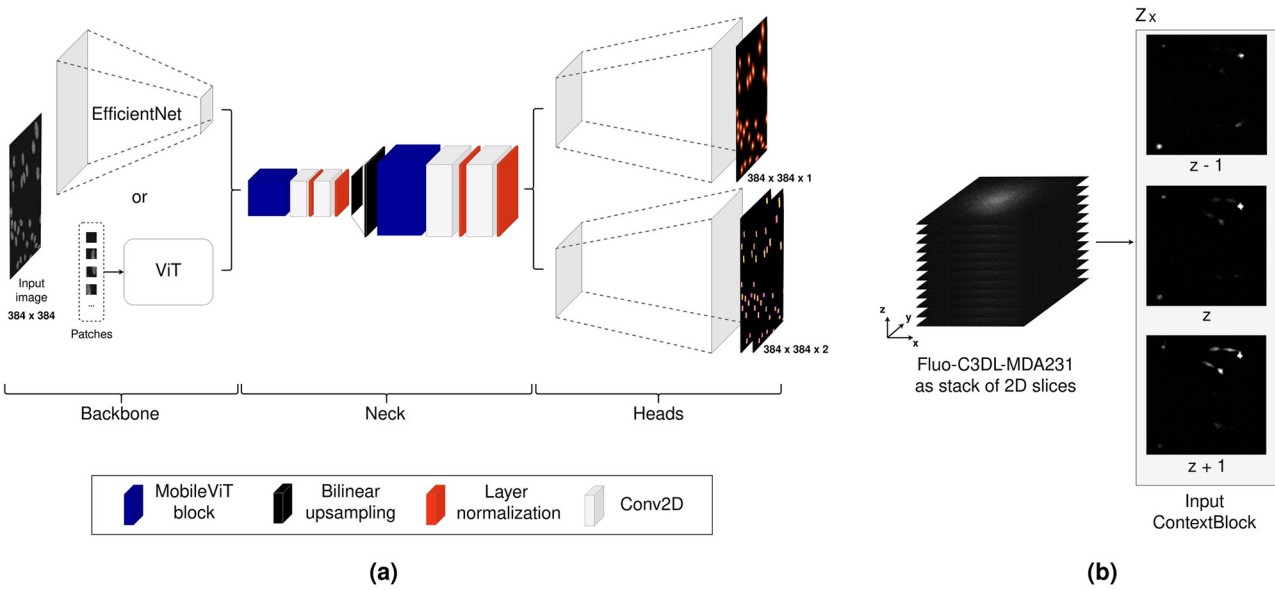

**Fig 2.** **(a)** CellCentroidFormer model. **Backbone:** Five blocks of an EfficientNetV2S [25] or a vision transformer (ViT). **Neck:** MobileViT blocks and convolutional layers. **Heads:** Fully convolutional upsampling blocks. **(b)** Adjacent z-slices as input for the ContextBlock.

$S(z+1)$ is merged in the third channel. In this way, context from adjacent slices can be highlighted, but the focus remains on the current slice. In general, prepending the Context-Block is a learned 2.5D approach for 3D data. For 2D data, additional features are extracted and 3 unique views of input images are generated.

## Decoding predictions

When decoding predictions, we first generate a binary map $E_{\text{blob}}$ that contains elliptical blobs by applying a threshold on the centroid heatmap $H_{\text{centroid}}$, as follows:

$$E_{\text{blob}}(x, y) = H_{\text{centroid}}(x, y) > 0.75 \tag{4}$$

Afterwards, we compute the image moments $M$ per blob and derive the corresponding centroid position $C$ using:

$$C\{x, y\} = \left\{ \frac{M_{10}}{M_{00}}, \frac{M_{01}}{M_{00}} \right\} \tag{5}$$

These centroid positions are then used to lookup the corresponding cell dimensions in the predicted cell height and cell width maps. In this work, we use the classic bounding box format of top left corner plus height and width as output.

## Results

### Datasets

We use publicly available datasets from the Cell Tracking Challenge [28] to evaluate our proposed method. In the first set of experiments, we are using the Fluo-N2DH-SIM+ dataset [29], which contains fluorescence microscopy images of simulated nuclei of HL60 cells. Fig 1 shows an example image of this dataset. In the second set of experiments, we combine the

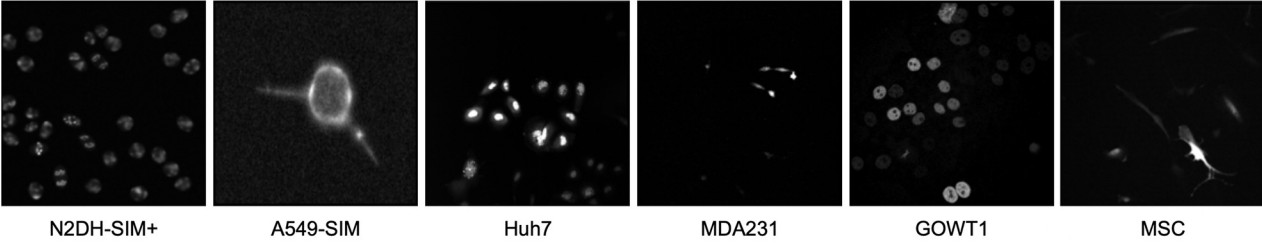

**Fig 3. Fluorescence microscopy datasets used for pre-training.**

Fluo-N2DH-SIM+ with the Fluo-C3DH-A549-SIM [30], the Fluo-C2DL-Huh7 [31], the Fluo-C3DL-MDA231, the Fluo-N2DH-GOWT1 [32], and the Fluo-C2DL-MSC datasets for pre-training. The Fluo-C3DH-A549-SIM contains 3D fluorescence microscopy images of simulated GFP-actin-stained A549 lung cancer cells. The Fluo-C2DL-Huh7 dataset contains 2D fluorescence microscopy images of human hepatocarcinoma-derived cells expressing the fusion protein YFP-TIA-1. The Fluo-C3DL-MDA231 dataset contains 3D fluorescence microscopy images of human breast carcinoma cells infected with a pMSCV vector including the GFP sequence, which are embedded in a collagen matrix. The Fluo-N2DH-GOWT1 dataset contains 2D fluorescence microscopy images of GFP-GOWT1 mouse stem cells. The Fluo-C2DL-MSC dataset contains 2D fluorescence microscopy images of rat mesenchymal stem cells on a flat polyacrylamide substrate. Fig 3 shows example images for all used datasets (2D slices for 3D datasets), in the order in which they are listed here. As pre-processing, we apply a median filter with a kernel size of 3 and perform min-max scaling. During training, we use the Fluo-N2DH-SIM+ and Fluo-C3DH-A549-SIM datasets that provide panoptic segmentation masks as ground truth, which we convert to bounding box annotations for cells.

## Comparing autoencoding schemes including pseudo-colorization and masking as pre-training on cell images

In this set of experiments, we are comparing different autoencoding schemes as pre-training for cell detection. We perform two types of training procedures, fine-tuning and head evaluation. For fine-tuning, we pre-train the models using different autoencoding schemes and afterwards allow all model parameters to adapt to the downstream task during training. For head evaluation, we pre-train the models using different autoencoding schemes and afterwards freeze the weights in the backbone and neck part of our model during training. Therefore, only the heads are adapting to the downstream task, which corresponds to 17% of the total model parameters.

**Dataset.** As dataset, we are using the training split of the Fluo-N2DH-SIM+ dataset. We perform geometric data augmentations such as elastic, perspective, shift, scale, and rotation transformations to increase the dataset size to 2150 samples. 80% of the resulting dataset are used for training and 20% for testing. The used dataset is rather small and we are comparing similar autoencoding schemes in this set of experiments. Therefore, we perform data augmentations beforehand instead of randomly and on-the-fly during training to train all models with the exact same samples and allow a fair comparison.

**Evaluation metrics.** As evaluation metrics, we use the structural similarity score (SSIM) [33], the mean intersection-over-union metric (mIoU), and bounding box average precision

metrics (AP). The structural similarity score measures the similarity of two matrices or gray-scale images and is used to measure the reconstruction quality during pre-training. For pseudo-colorized images we compute the SSIM score per color channel and average them. We do not use weight-sharing, but train both heads separately during pre-training. Therefore, we report the mean SSIM score of both model heads. The metric is defined as follows:

$$\text{SSIM}(x_1, x_2) = \frac{(2\mu_{x_1}\mu_{x_2} + (0.01L)^2)(2\sigma_{x_1 x_2} + (0.03L)^2)}{(\mu_{x_1}^2 + \mu_{x_2}^2 + (0.01L)^2)(\sigma_{x_1}^2 + \sigma_{x_2}^2 + (0.03L)^2)} \tag{6}$$

For two inputs, the mean $\mu$, the variance $\sigma$, and the dynamic range $L$ are computed. The mIoU metric is used to evaluate the centroid heatmap predictions. Matching our decoding procedure in Eq 4, we apply a threshold to the predicted and ground truth heatmap before computing the mIoU score. The metric is defined as:

$$\text{mIoU} = \frac{1}{C}\sum_C \frac{\text{TP}_C}{\text{TP}_C + \text{FP}_C + \text{FN}_C} \tag{7}$$

For the two class labels $C$, background and cell centroid blobs, the true positive (TP), false positive (FP), and false negative (FN) pixels are computed. The bounding box AP metrics are computed using the `pycocotools` software library, we refer to the COCO dataset [34] for more details.

**Experimental setup.** During pre-training, Adam [35] with its standard configuration is used as optimizer, the initial learning rate is set to $10^{-4}$, and reduced on plateaus by a factor of 10 to a minimum learning rate of $10^{-6}$. As pre-training loss, we are computing the pixel-wise mean squared error (MSE) between the reconstructed and the original microscopy images. All autoencoding schemes including pseudo-colorization are pre-trained for 75 epochs, whereas the basic autoencoding without pseudo-colorization is pre-trained for 50 epochs since it converges faster. Therefore, the basic autoencoding scheme has a lower relative total training time $T_{\text{rel}}$ than the remaining autoencoding schemes. During training, the same optimizer configuration and learning rate scheduling are used. As training loss, three Huber loss functions [36] are used, one loss function per output (heatmap, height, and width).

$$\mathcal{L}_{\text{Huber}}(y, \hat{y}) = \begin{cases} \frac{1}{2}(y - \hat{y})^2 & \text{if } y - \hat{y} \leq 1.0, \\ (y - \hat{y}) - \frac{1}{2} & \text{else} \end{cases} \tag{8}$$

The total loss is computed by a weighted sum of the three loss values:

$$\mathcal{L}_{\text{total}} = \mathcal{L}_{\text{heatmap}} + \frac{1}{2} \cdot \mathcal{L}_{\text{height}} + \frac{1}{2} \cdot \mathcal{L}_{\text{width}} \tag{9}$$

All models are trained for 50 epochs to detect cells. All training runs are completed three times, Table 1 shows the mean and standard deviation for all considered metrics.

**Results.** Basic autoencoding without pseudo-colorization yields the highest SSIM scores during pre-training, autoencoding using the `viridis` and `seismic` colormaps yields the second highest SSIM scores. This shows that these pre-training objectives are simpler than the remaining ones and therefore easier to learn. More difficult to learn seems autoencoding using the spectral `rainbow` colormap and most difficult are the two variations with and without masking using the pseudo-spectral `nipy_spectral` colormap. For all autoencoding schemes, the standard deviation from the SSIM score are low, showing that both model heads learn the objectives similarly well. In the fine-tuning training procedure, all autoencoding

**Table 1. Comparing autoencoding schemes including pseudo-colorization and masking as pre-training on cell images.**

| Pre-training | | | Training | | | | | | |
|---|---|---|---|---|---|---|---|---|---|
| Colormap | Masking | $SSIM_{test}^{heads}$ | $F_{DS}$ | $AP_{test}$ | $AP_{test}^{50}$ | $AP_{test}^{small}$ | $AP_{test}^{medium}$ | $mIoU_{train}^{heatmap}$ | $T_{rel}$ |
| - | - | - | 100% | 31.37 (1.27) | 71.93 (0.90) | 27.43 (1.08) | 44.37 (2.06) | 82.78 (0.82) | **1.0** |
| *Fine-tuning:* | | | | | | | | | |
| - | - | 0.9212 (0.0009) | 100% | 38.93 (2.60) | 78.87 (2.20) | 34.17 (2.40) | 54.33 (3.25) | 90.02 (2.04) | <u>1.9</u> |
| rainbow | - | 0.8392 (0.0003) | 100% | 40.60 (1.97) | 79.77 (1.00) | 35.53 (1.81) | 56.53 (2.65) | 89.79 (1.27) | 2.3 |
| viridis | - | 0.8790 (0.0087) | 100% | 40.70 (1.22) | 80.03 (0.85) | 35.97 (1.10) | 56.07 (1.37) | <u>91.21</u> (0.11) | 2.3 |
| seismic | - | 0.9020 (0.0035) | 100% | 40.33 (0.49) | 79.80 (0.35) | 35.73 (0.40) | 55.13 (1.46) | 90.41 (0.68) | 2.3 |
| nipy_spectral | - | 0.6302 (0.0004) | 100% | <u>43.10</u> (1.51) | <u>81.63</u> (0.90) | <u>38.07</u> (1.02) | <u>58.93</u> (2.93) | 90.95 (0.50) | 2.3 |
| nipy_spectral | ✓ | 0.6861 (0.0004) | 100% | **43.43** (1.90) | **81.90** (1.61) | **38.40** (2.15) | **59.27** (1.30) | **91.30** (0.55) | 2.3 |
| *Head evaluation:* | | | | | | | | | |
| - | - | 0.9212 (0.0009) | 100% | 22.30 (2.69) | 58.50 (4.77) | 19.30 (2.60) | 32.67 (2.57) | 69.74 (0.86) | <u>1.6</u> |
| nipy_spectral | - | 0.6302 (0.0004) | 100% | 30.87 (4.74) | <u>69.40</u> (4.42) | 27.30 (3.81) | 43.00 (6.81) | <u>73.41</u> (0.69) | 2.1 |
| nipy_spectral | ✓ | 0.6861 (0.0004) | 100% | **32.57** (1.88) | **70.17** (2.77) | **28.30** (1.66) | **47.17** (2.33) | **74.06** (0.34) | 2.1 |
| *Head evaluation with smaller training dataset:* | | | | | | | | | |
| - | - | 0.9212 (0.0009) | 20% | 9.45 (2.25) | 30.02 (5.91) | 7.00 (1.76) | 18.65 (3.39) | 66.69 (0.83) | <u>1.1</u> |
| nipy_spectral | - | 0.6302 (0.0004) | 20% | **16.80** (2.77) | **49.07** (5.00) | **14.57** (2.36) | <u>24.97</u> (4.26) | **72.29** (0.62) | 1.5 |
| nipy_spectral | ✓ | 0.6861 (0.0004) | 20% | <u>15.88</u> (3.47) | <u>44.65</u> (7.92) | <u>13.28</u> (3.08) | **25.78** (5.02) | <u>70.77</u> (1.17) | 1.5 |

As baseline, the first row shows a model trained from scratch with random weight initialization. Masking refers to our proposed padded masking scheme. Best scores per training procedure are bold, second best scores are underlined.

schemes achieve at least 10% higher values in all metrics than the baseline model trained from scratch. This demonstrates that autoencoding is generally an effective form of pre-training for this type of data. All cells in the Fluo-N3DH-SIM+ dataset fall into the small and medium large object categories with respect to COCO AP metrics. In general, medium sized cells are better detected, which is shown by higher AP scores. The basic autoencoding achieves on average about 2% lower AP scores than the very similar performing autoencoding schemes with the rainbow, viridis, and seismic colormaps. The two autoencoding schemes with the nipy_spectral colormap achieve AP scores that are about another 2.5% higher. When comparing these two autoencoding schemes, the autoencoding scheme that uses the proposed padded masking in addition to the nipy_spectral colormap achieves AP values that are consistently about 0.3% higher and thus performs best. The mIoU scores, which are achieved during training, are close to each other in this training procedure and within a range of 1.5%. In the following head evaluation procedure, the two previously best performing autoencoding schemes are compared with the basic autoencoding scheme. According to the significantly lower number of parameters, which can adapt to the downstream task, the performance drops significantly. On average, 10 to 20% lower AP scores are achieved. The performance differences between the three autoencoding schemes are comparable to those from the previous procedure. Therefore, the autoencoding scheme using the combination of nipy_spectral colormap and padded masking again performs best. This autoencoding scheme is also the only one that outperforms the differently trained baseline model with this training procedure. We increase the difficulty further by reducing the fraction of the training dataset $F_{DS}$ used to 20%. With this setup, the achieved AP scores are approximately halved. The two autoencoding schemes with the nipy_spectral colormap outperform the basic autoencoding again, whereas the autoencoding scheme with the nipy_spectral colormap but without masking

performs best in this setup. We hypothesize that the task switch from pure pseudo-colorization to cell detection is easier and therefore can be learned better with fewer training samples. The additional masking arguably enhances the learned representations, leading to better performance in the previous experimental setups. However, the associated reconstruction of masked image parts differs more than the pure pseudo-colorization from cell detection using centroid representations.

## Comparing self-supervised pre-training methods on cell images

In this set of experiments, we are comparing our proposed pre-training objective, pseudo-colorizing masked cells (PMC), with recent methods for self-supervised pre-training on cell images. Following [3], we pre-train our model using the contrastive learning framework SimCLR. In addition, we compare our method with edge-based self-supervision (EdgeSSV) [14] and pure masked image modeling with the masking scheme of masked autoencoders (MAE) [6]. The performance comparison is performed with backbone evaluation as training procedure. For this, the weights in the backbone are frozen after the pre-training, so that only the neck part and the two heads of our model adapt to the downstream task of cell detection.

**Experimental setup.** As described in the datasets Section, we pre-train with a diverse set of 6 fluorescence microscopy datasets. Subsequent training to evaluate performance is performed on the Fluo-C3DH-A549-SIM and the Fluo-N2DH-SIM+ datasets. The pre-training of all methods is performed for 75 epochs, the subsequent training for 50 epochs. During pre-training and training, we use the same optimizer setup and learning rate schedule as in Section 4.2 for all methods. During pre-training, we randomly perform geometric data augmentations on-the-fly such as flip, rotation, and crop transformations. An exception is the pre-training of SimCLR, where color affine transformations are performed in addition to geometric augmentations, as specified in the framework [9]. For color affine transformations, the prepended ContextBlock in our model and the corresponding input tenors with 3 channels additionally serve as adapters to apply such transformations to grayscale images. For SimCLR, we restrict the extent of the geometric augmentations, since the datasets used were created from time-lapse microscopy videos. In these videos, cells move only moderately between frames, so that successive frames are very similar. For contrastive learning, this means that strong geometric augmentations would lead to two differently augmented views of the same frame being more different from each other than from surrounding frames. However, since all frames except the two views of the same frame are used as negative samples in contrastive learning, we limit the extent of geometric transformations to only slight rotations (+/- 10˚) and cropping to a minimum of 90% of the original size. For SimCLR-pre-training, we replace the two model heads by one projection head with three fully connected layers, which have 256, 128, and 64 nodes. For the remaining methods, the architecture of our CellCentroidFormer model is not changed, only the top-layer is switched according to the output format between pre-training and training. On the Fluo-N2DH-SIM+ dataset, we additionally fine-tune a version of our method with a ViT backbone (PMC-ViT). We use a ViT-B/8 model with a patch size of $8 \times 8$ pixels and combine our pseudo-colorizing objective with vanilla MAE-like masked autoencoding as pre-training. All training runs are performed three times, we report mean and standard deviation in Table 2.

**Results.** As expected, all methods perform better on the Fluo-C3DH-A549-SIM dataset than on the Fluo-N2DH-SIM+ dataset. The Fluo-C3DH-A549-SIM dataset contains 3D data but only one cell is visible per frame. Furthermore, this dataset does not contain small cells, accordingly we can not compare $AP_{test}^{small}$ scores. The cells in this 3D dataset have an ellipsoidal shape, therefore their size in 2D slices decreases from the center of the volume to the border.

**Table 2. Comparing self-supervised pre-training methods on cell images.**

| Pre-training method | Det. dataset | $AP_{test}$ | $AP_{test}^{50}$ | $AP_{test}^{small}$ | $AP_{test}^{medium}$ | $AP_{test}^{large}$ | $mIoU_{train}^{heatmap}$ |
|---|---|---|---|---|---|---|---|
| SimCLR | A549-SIM | 24.40 (2.69) | 75.50 (0.99) | - (-) | 24.20 (2.12) | 27.10 (3.25) | 71.33 (1.21) |
| MAE-like | A549-SIM | 57.43 (2.23) | 86.13 (1.75) | - (-) | 49.83 (2.71) | **71.30** (2.75) | **94.30** (1.08) |
| EdgeSSV | A549-SIM | 55.70 (3.11) | 87.35 (2.90) | - (-) | 47.10 (3.25) | 70.40 (2.97) | 93.04 (0.15) |
| PMC (Ours) | A549-SIM | **57.70** (3.54) | **87.90** (1.41) | - (-) | **51.80** (2.97) | 70.00 (3.11) | 93.08 (0.80) |
| SimCLR | N2DH-SIM+ | 24.10 (0.99) | 60.55 (1.06) | 20.10 (0.99) | 37.45 (0.92) | - (-) | 68.10 (1.72) |
| MAE-like | N2DH-SIM+ | 27.25 (1.34) | 66.55 (1.63) | 23.40 (1.13) | 39.85 (1.77) | - (-) | 84.78 (1.44) |
| EdgeSSV | N2DH-SIM+ | 31.30 (0.71) | 72.50 (1.41) | 26.95 (0.64) | 45.20 (0.57) | - (-) | **86.23** (1.29) |
| PMC (Ours) | N2DH-SIM+ | 33.75 (0.64) | 73.10 (0.57) | 28.90 (0.85) | 49.25 (0.64) | - (-) | 85.15 (1.06) |
| PMC-ViT (Ours) | N2DH-SIM+ | **51.60** (0.26) | **87.07** (1.10) | **46.27** (0.67) | **64.37** (0.21) | - (-) | 78.58 (1.97) |

Best scores are bold, second best scores are underlined. A549-SIM and N2DH-SIM+ refer to the Fluo-C3DH-A549-SIM and Fluo-N2DH-SIM+ datasets.

Accordingly, the detection in 2D slices from the center of the volume is evaluated with the $AP_{test}^{large}$ metric, and towards the border with the $AP_{test}^{medium}$ metric. In the inner 2D slices, cells are detected more accurately by all methods. On the Fluo-N2DH-SIM+ dataset, medium sized cells are detected more accurately than small cells by all methods. On both datasets, SimCLR performs worst overall, which can be explained by the fact that the similarity between individual frames in time-lapse videos makes training for SimCLR more challenging. MAE-like masking performs significantly worse on the Fluo-N2DH-SIM+ dataset than on the Fluo-C3DH-A549-SIM dataset. This supports our hypothesis that this masking scheme is better suited for data with one concept (one cell) per image. On the Fluo-C3DH-A549-SIM dataset, MAE-like masking performs second best and yields $AP_{test}$ scores within 1% of our proposed method. The EdgeSSV method achieves high AP scores on both datasets. Overall, our proposed method achieves the highest AP scores on both datasets. However, considering that on the Fluo-C3DH-A549-SIM dataset the three best methods yield $AP_{test}$ scores within a range of 2% and the higher standard deviation values, the performance difference is less significant for this dataset. On the Fluo-N2DH-SSIM+ dataset, our method with a larger ViT backbone achieves the highest AP scores. This shows that our method scales well with model size and can be used for training larger models on small datasets. Interestingly, our method does not achieve the highest $mIoU_{train}^{heatmap}$ scores in either dataset, which is measured on the training data. This suggests that our method generalizes better and other methods tend to overfit somewhat more on the training data.

## Discussion

Our experiments reveal that pseudo-colorization is an effective extension to autoencoding for pre-training on cell images. Pseudo-color autoencoding with pseudo-spectral colormaps, which approximate the semantics of cell images, yields the best results on fluorescence microscopy images. As recently shown for natural images [37], the combination of standalone self-supervision objectives with masked image modeling can also further improve performance on biomedical cell images. The proposed unique combination of pseudo-color autoencoding and masked image modeling for cell images can outperform several autoencoding schemes, contrastive learning, and edge-based self-supervision. Our proposed masking scheme consists of smaller patches and covers a much smaller area of images than the typical masking scheme of masked autoencoders. This prevents object instances from being completely masked. We found this to be beneficial for subsequent training on object detection in our experiments. Our

CellCentroidFormer model with a ViT backbone achieves the computational efficiency of masked autoencoders since only unmasked patches are processed in the backbone during pre-training. Our CellCentroidFormer model with an EfficientNet backbone has no special mechanism to suppress the artificial edges added by masking. Nevertheless, as shown in Fig 1(c), no edge artifacts are visible in the reconstructions. We hypothesize that the self-attention layers in the neck part of our model help to suppress these edge artifacts, since self-attention can be understood as a form of generalized spatial smoothing [38]. However, it is plausible that our method can be further improved by using masked convolutions [12] instead of regular convolutions to suppress artificial edges. To this end, we show that our proposed pre-training method can achieve good performance on small cell image datasets using an autoencoder-like model without special architectural modifications.

## Acknowledgments

We acknowledge support by the KIT-Publication Fund of the Karlsruhe Institute of Technology.

## Author Contributions

**Conceptualization:** Royden Wagner.

**Methodology:** Royden Wagner.

**Software:** Royden Wagner.

**Validation:** Royden Wagner.

**Visualization:** Royden Wagner.

**Writing – original draft:** Royden Wagner.

**Writing – review & editing:** Carlos Fernandez Lopez, Christoph Stiller.

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
