## [Decision Letter · Decision Letter 0]

2 Jun 2023

PONE-D-23-10029Self-supervised Pseudo-colorizing of Masked CellsPLOS ONE

Dear Dr. Wagner,

Thank you for submitting your manuscript to PLOS ONE. After careful consideration, we feel that it has merit but does not fully meet PLOS ONE’s publication criteria as it currently stands. Therefore, we invite you to submit a revised version of the manuscript that addresses the points raised during the review process.

We look forward to receiving your revised manuscript.

Kind regards,

Vijayalakshmi G V Mahesh, Ph.D

Academic Editor

PLOS ONE

2. Please remove your figures from within your manuscript file, leaving only the individual TIFF/EPS image files, uploaded separately. These will be automatically included in the reviewers’ PDF

Reviewers' comments:

Reviewer's Responses to Questions

**Comments to the Author**

1. Is the manuscript technically sound, and do the data support the conclusions?

Reviewer #1: Yes

Reviewer #2: Yes

2. Has the statistical analysis been performed appropriately and rigorously? 

Reviewer #1: Yes

Reviewer #2: Yes

3. Have the authors made all data underlying the findings in their manuscript fully available?

Reviewer #1: Yes

Reviewer #2: Yes

4. Is the manuscript presented in an intelligible fashion and written in standard English?

Reviewer #1: Yes

Reviewer #2: Yes

5. Review Comments to the Author

Reviewer #1: The manuscript is organised and presented in an intelligible fashion and technically sounds good.The statistical analysis is performed rigorously and the manuscript.The manuscript can be recommended for publication

Reviewer #2: 1. Summary of the paper

The paper proposes self supervised learning for analysis of cells in biomedical microscopy images. Cell images are analyzed using a combination of pseudo coloring and masked image modeling (MIM). Four color maps rainbow, seismic, nipy-spectral and viridis color manps are used for pseudo coloring based on the semantics of the data. Cell analysis has wide variety of applications like conduction of viability studies on cell population, processing genetic material within a cell, visualization of cellular structures to name a few.

Experiments are conducted on synthetic Fluo-N2DH-SIM+ and other datasets. Data augmentation is used to increase the sample size.

There are various datasets used and the reason behind using data augmentation is not clear

The proposed method is evaluated using SSIM, MIoU, and AP.

2. Strengths and weakness of the paper

The proposed method is a unique combination of pseudo-coloring and masked image modeling and might outperform contrastive learning.

3. Any additional comments on the submission

The computational complexity of the proposed method seems high.

Decision: Accept

6. PLOS authors have the option to publish the peer review history of their article (what does this mean?). If published, this will include your full peer review and any attached files.

Reviewer #1: No

Reviewer #2: **Yes: **Dr. Surabhi Narayan

---

## [Author Response · Author response to Decision Letter 0]

12 Jul 2023

Reviewer 1: Thank you for your feedback.

Reviewer 2: Thank you for your feedback. Below are our responses to the points you

raised during the review.

P: There are various datasets used and the reason behind using data augmentation is

not clear.

R: We use only the simulated datasets for evaluation to use a perfect ground truth,

since the other datasets do not contain perfect annotations, but mainly silver truth

annotations generated from previous challenge submissions (see:

http://celltrackingchallenge.net/annotations). However, we pre-compute the

augmented samples to train all methods on the same data and make the comparison as

fair as possible.

P: The computational complexity of the proposed method seems high.

R: We assume that you are referring to the higher computational complexity of our

method compared to a vanilla masked autoencoder (MAE). MAE has a lower

computational complexity since the vision transfomer (ViT) used as encoder is only

applied to non-masked patches whereas the EfficientNet backbone in our model is

applied to both masked and non-masked patches. Therefore, we additionally

implemented a version of our CellCentroidFormer model with a ViT backbone to

combine vanilla masked autoencoding with pseudo-colorizing and achieve the same

computational efficiency. We added the results to Table 2, adjusted Fig 2, and published

the code at: https://github.com/roydenwa/pseudo-colorize-masked-cells

---

## [Decision Letter · Decision Letter 1]

10 Aug 2023

Self-supervised pseudo-colorizing of masked cells

PONE-D-23-10029R1

Dear Dr. Wagner,

We’re pleased to inform you that your manuscript has been judged scientifically suitable for publication and will be formally accepted for publication once it meets all outstanding technical requirements.

Kind regards,

Vijayalakshmi G V Mahesh, Ph.D

Academic Editor

PLOS ONE

Additional Editor Comments (optional):

Reviewers' comments:

Reviewer's Responses to Questions

**Comments to the Author**

1. If the authors have adequately addressed your comments raised in a previous round of review and you feel that this manuscript is now acceptable for publication, you may indicate that here to bypass the “Comments to the Author” section, enter your conflict of interest statement in the “Confidential to Editor” section, and submit your "Accept" recommendation.

Reviewer #1: All comments have been addressed

Reviewer #2: All comments have been addressed

2. Is the manuscript technically sound, and do the data support the conclusions?

Reviewer #1: Yes

Reviewer #2: Yes

3. Has the statistical analysis been performed appropriately and rigorously? 

Reviewer #1: Yes

Reviewer #2: Yes

4. Have the authors made all data underlying the findings in their manuscript fully available?

Reviewer #1: Yes

Reviewer #2: Yes

5. Is the manuscript presented in an intelligible fashion and written in standard English?

Reviewer #1: Yes

Reviewer #2: Yes

6. Review Comments to the Author

Reviewer #1: the minor revisions has been addressed and the manuscript shall be recommended for publication

Reviewer #2: The queries are sufficiently answered by the authors. The authors have clarified the query about data augmentation and the use of synthetic data set.

7. PLOS authors have the option to publish the peer review history of their article (what does this mean?). If published, this will include your full peer review and any attached files.

Reviewer #1: **Yes: **Chandra prabha R

Reviewer #2: No

---

## [Editor Report · Acceptance letter]

15 Aug 2023

PONE-D-23-10029R1 

Self-supervised pseudo-colorizing of masked cells  

Dear Dr. Wagner:

I'm pleased to inform you that your manuscript has been deemed suitable for publication in PLOS ONE. Congratulations! Your manuscript is now with our production department. 

Kind regards, 

on behalf of

Dr. Vijayalakshmi G V Mahesh 

Academic Editor

PLOS ONE